# Assessing the safety of home oximetry for COVID-19: a multisite retrospective observational study

Jonathan Clarke ![ORCID],[1] Kelsey Flott,[2] Roberto Fernandez Crespo,[2] Hutan Ashrafian ![ORCID],[2] Gianluca Fontana,[2] Jonathan Benger ![ORCID],[3] Ara Darzi,[2] Sarah Elkin[4]

[1]Centre for Mathematics of Precision Healthcare, Imperial College London, London, UK
[2]Institute of Global Health Innovation, Imperial College London, London, UK
[3]NHS Digital, Leeds, UK
[4]National Heart and Lung Institute, Imperial College London, London, UK

**Correspondence to**
Dr Jonathan Clarke;
j.clarke@imperial.ac.uk

## ABSTRACT

**Objectives** To determine the safety and effectiveness of home oximetry monitoring pathways for patients with COVID-19 in the English National Health Service.

**Design** Retrospective, multisite, observational study of home oximetry monitoring for patients with suspected or proven COVID-19.

**Setting** This study analysed patient data from four COVID-19 home oximetry pilot sites in England across primary and secondary care settings.

**Participants** A total of 1338 participants were enrolled in a home oximetry programme across four pilot sites. Participants were excluded if primary care data and oxygen saturations at rest at enrolment were not available. Data from 908 participants were included in the analysis.

**Interventions** Home oximetry monitoring was provided to participants with a known or suspected diagnosis of COVID-19. Participants were enrolled following attendance to emergency departments, hospital admission or referral through primary care services.

**Results** Of 908 patients enrolled into four different COVID-19 home oximetry programmes in England, 771 (84.9%) had oxygen saturations at rest of 95% or more, and 320 (35.2%) were under 65 years of age and without comorbidities. 52 (5.7%) presented to hospital and 28 (3.1%) died following enrolment, of which 14 (50%) had COVID-19 as a named cause of death. All-cause mortality was significantly higher in patients enrolled after admission to hospital (OR 8.70 (2.53–29.89)), compared with those enrolled in primary care. Patients enrolled after hospital discharge (OR 0.31 (0.15–0.68)) or emergency department presentation (OR 0.42 (0.20–0.89)) were significantly less likely to present to hospital than those enrolled in primary care.

**Conclusions** This study finds that home oximetry monitoring can be a safe pathway for patients with COVID-19; and indicates increases in risk to vulnerable groups and patients with oxygen saturations <95% at enrolment, and in those enrolled on discharge from hospital. Findings from this evaluation have contributed to the national implementation of home oximetry across England.

## STRENGTHS AND LIMITATIONS OF THIS STUDY

⇒ This multisite study examines outcomes and variation between four COVID-19 home oximetry remote monitoring pathways in England.
⇒ It uses linked data from primary and secondary care, alongside mortality data and data collected from the oximetry pathway to understand clinical outcomes and prior risk factors.
⇒ It examines the clinical status of patients enrolled under different home oximetry pathways and examines their rates of hospital attendance and all-cause mortality.
⇒ The study is limited to four sites, and is unable to comment on the outcomes of patients not enrolled into these pathways or on the performance of pathways not included in the study.

existed regarding the role of telemedicine and digital technologies in restructuring how healthcare is delivered, indicating an opportunity to expand the use of virtual pathways.[1] Such evidence suggests that care delivered remotely can, in many circumstances, safely meet patients' clinical needs and personal preferences.[1–3] Specifically, remote monitoring pathways, those that rely on an initial point of contact with the health services followed by continuous monitoring via phone calls, digital or app-based diaries or wearable sensors, have also demonstrated effectiveness, especially when supported by behavioural change models.[4–6] However, the clinical effectiveness, safety and economic utility of this type of monitoring are context dependent and vary considerably across clinical conditions; therefore, more evidence is required to fully assess their impact.[4]

During the COVID-19 pandemic, the UK National Health Service (NHS), and health systems across the world, rapidly adopted novel remote monitoring pathways, many relying on home pulse oximetry.[7–9] These pathways were often initiated quickly in an

## INTRODUCTION
Before the outbreak of the COVID-19 pandemic, compelling evidence already

attempt to proactively manage patients without risking unnecessary patient travel to hospital, but subsequently varied in terms of the devices used, patients included and the method of implementation. The available literature suggests potential for home management of COVID-19 to support a positive patient experience.[10–12] However, while pulse oximetry and trends over time appear to be an effective way of detecting deterioration, the evidence surrounding the safety of oximetry devices, specifically low-cost pocket oximeters, is variable and more research is required to understand what oxygen saturation thresholds should trigger a patient to seek in-person care.[13 14] Overall, the literature surrounding whether home oximetry monitoring for patients with COVID-19 is safe remains inconclusive.[9 15]

In the UK, NHS England/Improvement, the body responsible for improved delivery of NHS care, in partnership with NHS Digital and Imperial College London, set out to understand, quantitatively, whether home oximetry monitoring was a safe clinical pathway to implement nationally. Following the first UK peak of COVID-19, in Spring 2020, pilots of home oximetry monitoring tested a system-wide approach to the early detection of COVID-19 in the community. As part of this pilot, a rapid evaluation was conducted to determine whether home oximetry monitoring was a safe clinical pathway. Based on existing literature, the initial hypothesis was that early recognition, escalation, admission and treatment could save lives in COVID-19 and would be a safe approach to adopt. In December 2020, informed by the findings of this study, NHS England and NHS Improvement recommended that all Clinical Commissioning Groups put in place a COVID-19 home oximetry service. At the time of writing, more than 20 000 patients in England have been treated on similar pathways to those included in this study.[16]

## Aims and objectives

This study aimed to answer the research question: *are home oximetry monitoring pathways safe and effective for patients with COVID-19 in the English NHS?*

The primary objective was to determine whether patients suffered any adverse consequences as a result of home oximetry monitoring. The secondary objective was to explore whether the recommendations relating to oxygen saturation that were published in June 2020 in the NHS COVID-19 assessment pathway were correct. The final objective was to contribute to a recommendation to NHS England/Improvement regarding the safety and suitability of home oximetry monitoring as a national COVID-19 pathway.

## METHODS

### Setting and design

To answer this question, Imperial College NHS Healthcare Trust, National Institute for Health Research Imperial Patient Safety Translational Research Centre at the Institute of Global Health Innovation (IGHI) at Imperial College London, working in partnership with NHS Digital, assessed existing evidence concerning home oximetry monitoring for respiratory conditions and analysed prospective data collected from four sites.

This was a retrospective, multisite, observational study of home oximetry monitoring for patients with suspected or proven COVID-19 in England during Summer 2020, including an analysis of patient data from four pilot sites: North West London, Slough, South Tees and Watford. Sites varied in how patients were enrolled onto pathways; from primary care, after a presentation to the emergency department or following a hospital admission, or a combination of these. In each site, patients were enrolled onto the oximetry pathway and were provided with a pulse oximeter to measure their oxygen saturations and heart rate over time. These recordings, in addition to a patient's symptoms and perceived overall well-being, were communicated to their nominated healthcare professional (either a general practitioner, practice nurse or hospital doctor) through a combination of app-based platforms and paper diaries. The frequency of recording varied between pathways, from daily to several times per day. In the case of clinical deterioration or other concerns, patients were able to contact their nominated healthcare professional for advice and escalation of care could be initiated as appropriate. While services may vary between sites, the current standard operating procedure for COVID-19 home oximetry services in England can be accessed on the NHS Digital website.[17]

Quantitative analysis was conducted in the IGHI's Big Data and Analytical Unit (BDAU). The BDAU provides a fully certified ISO 27001:2013 research environment within Imperial College and is fully compliant with NHS IG Toolkit Level 3 (EE133887). Analysis took place within the Imperial College London BDAU.

### Information and data

Four separate data sets linked by a pseudonymised patient identifier were provided by NHS Digital for the express purpose of this evaluation. A full list of variables collected is included in online supplemental file 1 and summarised below.

### Primary care records

Data pertaining to patient demographics and clinical comorbidities were obtained from the General Practice Extraction Service Data for Pandemic Planning and Research (GDPPR) for all patients enrolled into a home oximetry monitoring programme at one of the four sites. Records were available from the start of a patient's primary care record in the practice to the date of data transfer (25 September 2020).

### Home oximetry monitoring records

For each patient enrolled into a home oximetry monitoring programme at the four sites, data were collected from individual home oximetry monitoring providers consisting of the date of enrolment, oxygen saturations at

rest at enrolment and the clinical pathway to which they were enrolled.

## Hospital records

The dates, outcomes of accident and emergency (A&E) presentations and admissions to hospital for patients while enrolled in the home oximetry programme were obtained from case note review by participating sites. Data were returned by sites in mid-September 2020.

## Mortality records

The date and causes (as International Classification of Diseases 10th Revision (ICD-10) codes) of death were provided for each patient whose death had been recorded by the Office for National Statistics after enrolment to the home oximetry programme until the date of data transfer (25 September 2020).

## Analysis and statistical procedure

Data were linked according to the pseudonymised patient identifier. A single cohort of patients was identified where GDPPR data and oxygen saturations at rest at enrolment were available. The demographic and clinical characteristics of this population were described and differences in patient and clinical characteristics were compared between routes of enrolment using pairwise Fisher's exact tests for count data and Mann-Whitney U (MWU) tests for non-parametric continuous data.

The likelihood of a patient presenting to hospital at least once following enrolment to the home oximetry monitoring programme was examined using univariable and multivariable logistic regression models. The likelihood of all-cause mortality following enrolment to the home oximetry monitoring programme was also examined using univariable and multivariable logistic regression models. Frequent absence of dates of enrolment or dates of hospital admission precluded formal temporal evaluation of time to hospital presentation or mortality.

## RESULTS

A total of 1338 patients were recorded as being enrolled into the home oximetry programme at one of the four pilot sites. Of these, GDPPR records were available for 1242 patients (92.8%). Of these patients, a recorded oxygen saturation level at enrolment onto the home oximetry was present for 908 patients (73.1%).

## Comparison of enrolment pathways

The characteristics of the included population, overall and according to the route of enrolment, are described in table 1. Three hundred and two patients were enrolled from primary care (33.4%), 342 from A&E (37.9%) and 259 following discharge from hospital (28.9%). Route of enrolment was missing for five patients (0.6%).

Oxygen saturations at enrolment were significantly lower in those enrolled on discharge from hospital (96%) than through A&E (97%) or primary care (98%) (MWU, p<0.001). No difference was observed between primary care and A&E pathways (MWU, p=0.085). Similarly, there was no significant difference in the proportion of patients enrolled with oxygen saturations of 95% or more in primary care (92.0%) or A&E (91.8%) (Fisher's exact test, p=1.000); however, only 67.6% of patients enrolled after hospital discharge had oxygen saturations of 95% or more (pairwise Fisher's exact tests, p<0.001).

Patients enrolled through A&E were most likely to have no comorbidities (47.1%) compared with primary care (37.4%) and hospital discharge (27.0%). Primary care patients were less likely to have comorbidities than those enrolled after hospital discharge (pairwise Fisher's exact tests, p<0.05 in all cases). Patients enrolled through A&E were younger (median age=50 years) than those enrolled in primary care (55 years) and on hospital discharge (63 years) (MWU, p<0.001 in all cases).

## Presentation to hospital

A total of 69 presentations to hospital were made by 52 patients (5.7%) after enrolment. Forty of these patients (76.9%) were admitted to hospital, and eight patients (15.4%) presented more than once. The proportion of patients presenting to hospital according to age group and number of clinical comorbidities is shown in figures 1 and 2.

3.1% of patients under the age of 65 years and without comorbidities presented to hospital, compared with 5.0% of those under 65 with comorbidities and 9.7% of those aged 65 and over. Similarly, 5.3% of patients with oxygen saturations of 95% or more at enrolment presented to hospital, compared with 6.1% of those with oxygen saturations of 93%–94% and 10.9% of those with oxygen saturations less than 93% at enrolment (table 2).

In univariable logistic regression models, presentation to hospital was significantly more likely in older patients and those with more clinical comorbidities, while those initiated on the home oximetry through A&E or on discharge from hospital were less likely to re-present to hospital (table 3). In the multivariable model, increasing age (OR 1.03, p=0.018) was associated with significantly higher odds of presentation to hospital, while those initiated through A&E (OR 0.42, p=0.024) and following discharge from hospital (OR 0.31, p=0.003) were significantly less likely to present to hospital (table 3).

## All-cause mortality

A total of 28 patients (3.1%) died of any cause following enrolment. Fourteen of these patients (50.0%) had COVID-19 as a named cause of death (ICD-10 code U07.1), and 12 (42.9%) had COVID-19 as the underlying cause of death. All-cause mortality by age group and number of clinical comorbidities is shown in figures 1 and 2. Only patients with one or more comorbidities died, while 17 (60.1%) patients who died had four or more clinical comorbidities.

None of the 320 patients under the age of 65 and without comorbidities included in the study died during the study period, compared with 0.9% of those under 65

**Table 1** Characteristics of the study population, overall and according to the route of enrolment

| | | Primary care | Accident and emergency | Inpatient discharge | Overall |
|---|---|---|---|---|---|
| Total patients | | 302 | 342 | 259 | 908 |
| Female | | 180 (59.6) | 205 (59.9) | 120 (46.3) | 508 (55.9) |
| Median age (years) | | 55 | 50 | 63 | 54 |
| Ethnicity | White | 113 (37.4) | 160 (46.8) | 140 (54.1) | 415 (45.7) |
| | Black, Asian and minority Ethnic | 82 (27.2) | 103 (30.1) | 59 (22.8) | 244 (26.9) |
| | Not recorded | 107 (35.4) | 79 (23.1) | 60 (23.2) | 249 (27.4) |
| Overweight or obese | | 108 (35.8) | 189 (55.3) | 160 (61.8) | 458 (50.4) |
| Median oxygen saturations | | 98 | 97 | 96 | 97 |
| Clinical severity based on oxygen saturations | Mild (≥95%) | 278 (92.1) | 314 (91.8) | 175 (67.6) | 771 (84.9) |
| | Moderate (93%–94%) | 17 (5.6) | 22 (6.4) | 42 (16.2) | 82 (9.0) |
| | Severe (≤92%) | 7 (2.3) | 6 (1.8) | 42 (16.2) | 55 (6.1) |
| Clinical comorbidities (n) | 0 | 113 (37.4) | 161 (47.1) | 70 (27.0) | 346 (38.1) |
| | 1 | 75 (24.8) | 105 (30.7) | 70 (27.0) | 252 (27.7) |
| | 2 | 42 (13.9) | 45 (13.2) | 44 (17.0) | 131 (14.4) |
| | 3+ | 72 (23.8) | 31 (9.1) | 75 (29.0) | 179 (19.7) |
| Hypertension | | 102 (33.8) | 65 (19.0) | 111 (42.9) | 278 (30.8) |
| Depression | | 86 (28.5) | 84 (24.6) | 72 (27.8) | 242 (26.8) |
| Asthma | | 75 (24.8) | 83 (24.3) | 49 (18.9) | 207 (22.9) |
| Steroid use | | 87 (28.8) | 62 (18.1) | 54 (20.8) | 203 (22.5) |
| Diabetes mellitus | | 56 (18.5) | 43 (12.6) | 63 (24.3) | 162 (17.9) |
| Pregnancy | | 37 (12.3) | 74 (21.6) | 23 (8.9) | 134 (14.8) |
| Chronic heart disease | | 43 (14.2) | 35 (10.2) | 55 (21.2) | 133 (14.7) |
| Mild frailty | | 41 (13.6) | 17 (5.0) | 42 (16.2) | 100 (11.1) |
| Moderate frailty | | 45 (14.9) | 8 (2.3) | 43 (16.6) | 96 (10.6) |
| Chronic respiratory disease | | 39 (12.9) | 17 (5.0) | 35 (13.5) | 91 (10.1) |
| Chronic kidney disease | | 32 (10.6) | 17 (5.0) | 41 (15.8) | 90 (10.0) |
| Cancer | | 32 (10.6) | 23 (6.7) | 23 (8.9) | 78 (8.6) |
| Hypothyroidism | | 29 (9.6) | 22 (6.4) | 17 (6.6) | 68 (7.5) |
| Hypercholesterolaemia | | 30 (9.9) | 14 (4.1) | 22 (8.5) | 66 (7.3) |
| Chronic neurological disease | | 28 (9.3) | 7 (2.0) | 25 (9.7) | 60 (6.6) |
| Severe frailty | | * | * | * | 53 (5.9) |
| Stroke | | * | * | * | 46 (5.1) |
| Immunosuppression drug | | 8 (2.6) | 12 (3.5) | 7 (2.7) | 27 (3.0) |
| Dementia | | * | * | * | 25 (2.8) |
| Mental illness | | 8 (2.6) | 6 (1.8) | 8 (3.1) | 22 (2.4) |
| Epilepsy | | * | * | * | 20 (2.2) |
| Peripheral vascular disease | | * | * | * | 13 (1.4) |
| Autoimmune disease | | * | * | * | 13 (1.4) |
| Chronic liver disease | | * | * | * | 9 (1.0) |
| Learning disability | | * | * | * | 6 (0.7) |

Route of enrolment was missing for 5 of 908 patients. Comorbidity case numbers are shown with an asterisk (*) for routes of enrolment where one or more routes of enrolment involved 5 or fewer patients.

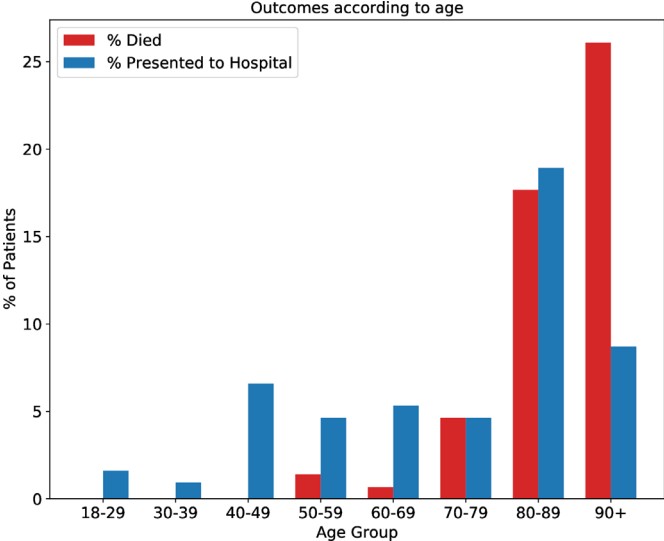

**Figure 1** Percentage all-cause mortality and presentation to hospital according to patient age.

with comorbidities and 9.3% of those aged 65 and over. 2.6% of patients with oxygen saturations of 95% or more at enrolment died during the study period, compared with 6.1% of those with oxygen saturations of 93%–94% and 5.5% of those with oxygen saturations less than 93% at enrolment (table 2).

In univariable logistic regression models, all-cause mortality was significantly more likely in older patients, those of black, Asian and minority ethnic ethnicity, with more clinical comorbidities, who were overweight or obese and those initiated on the home oximetry following discharge from hospital (table 4). In the multivariable model, increasing age (OR 1.08, p < 0.001), more clinical comorbidities (OR 1.45, p=0.009), being overweight or obese (OR 4.83, p=0.002) and being initiated on the home oximetry following discharge from hospital (OR

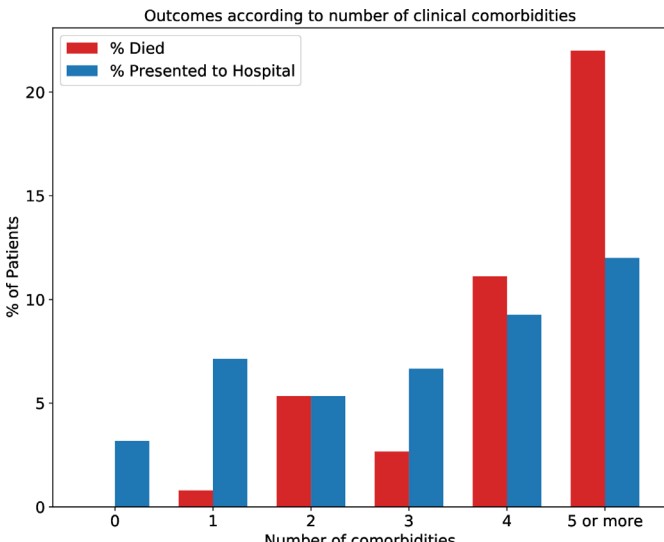

**Figure 2** Percentage all-cause mortality and presentation to hospital according to number of clinical comorbidities.

8.70, p=0.001) were associated with significantly higher odds of all-cause mortality (table 4).

## DISCUSSION
### Statement of key findings
In terms of clinical safety, only 5.7% of patients presented to hospital after enrolment. The likelihood of presenting to hospital increased with age, but was not significantly associated with ethnicity, number of clinical comorbidities, obesity or oxygen saturations at enrolment. However, all-cause mortality was significantly more likely with increasing age, number of clinical comorbidities and obesity. All-cause mortality was also higher for those patients who initiated the pathway after discharge from hospital. Finally, most patients using home oximetry monitoring were of low clinical severity on initiation and neither died nor presented to hospital during the study period. Therefore, the COVID-19 assessment pathway published in June 2020 is clinically appropriate, and its recommendations regarding oxygen saturation thresholds are correct.

While hospital presentation and mortality were associated with lower oxygen saturations at enrolment, this was not a statistically significant relationship after controlling for other patient factors. Therefore, although oxygen saturation at enrolment may provide an important tool by which to stratify clinical severity, such assessments must also incorporate a wider understanding of factors determining outcome including age, obesity and clinical comorbidities.

None of the 320 participants aged less than 65 and without comorbidities died during the study period. These individuals accounted for 33.3% of all participants which suggests that during the pilots a large proportion of the participants were low risk at enrolment and did not suffer any adverse outcome. Wider implementation should therefore focus on higher risk groups to ensure efficient use of limited health and care resources.

Statistically significant variation in the characteristics of patients, clinical severity of illness, rates of hospital presentation and all-cause mortality were observed between the three routes of enrolment. Patients enrolled following a hospital admission had 8.7 times higher odds of all-cause mortality than patients enrolled in primary care after adjusting for age, sex, ethnicity, obesity, clinical comorbidities and oxygen saturation at enrolment. Conversely, compared with those enrolled in primary care, patients enrolled after an A&E presentation or hospital admission had significantly lower odds of further presentation to hospital after enrolment. Collectively, this indicates differences in patient characteristics and the risk of adverse events according to the route of enrolment. Those enrolled in primary care are generally the least unwell and go on to have lower rates of hospital presentation and mortality. The finding that patients enrolled in A&E or following hospital admission have lower rates of hospital presentation may indicate reluctance on the part

**Table 2** Number of patients, frequency of hospital presentation and all-cause mortality according to risk groups and oxygen saturations at enrolment

| Outcome | Risk group | Oxygen saturation severity | | | |
| | | Mild (≥95%) | Moderate (93%–94%) | Severe (≤92%) | Overall |
| --- | --- | --- | --- | --- | --- |
| Total patients | ≥65 years | 203 (75.5) | 37 (13.8) | 29 (10.8) | 269 (29.6) |
| | <65 years with comorbidities | 277 (86.8) | 29 (9.1) | 13 (4.1) | 319 (35.1) |
| | <65 years without comorbidities | 291 (90.9) | 16 (5.0) | 13 (4.1) | 320 (35.2) |
| | Overall | 771 (84.9) | 82 (9.0) | 55 (6.1) | 908 (100.0) |
| Hospital presentation | ≥65 years | 18 (8.9) | 3 (8.1) | 5 (17.2) | 26 (9.7) |
| | <65 years with comorbidities | 13 (4.7) | 2 (6.9) | 1 (7.7) | 16 (5.0) |
| | <65 years without comorbidities | 10 (3.4) | 0 (0.0) | 0 (0.0) | 10 (3.1) |
| | Overall | 41 (5.3) | 5 (6.1) | 6 (10.9) | 52 (5.7) |
| All-cause mortality | ≥65 years | 18 (8.9) | 4 (10.8) | 3 (10.3) | 25 (9.3) |
| | <65 years with comorbidities | 2 (0.7) | 1 (3.4) | 0 (0.0) | 3 (0.9) |
| | <65 years without comorbidities | 0 (0.0) | 0 (0.0) | 0 (0.0) | 0 (0.0) |
| | Overall | 20 (2.6) | 5 (6.1) | 3 (5.5) | 28 (3.1) |

of patients to return to hospital during the same period of illness. Patients enrolled following hospital discharge appear to be at particularly high risk of mortality and may therefore represent a patient population that should either be offered more intensive monitoring or alternative approaches to reducing the risk of subsequent mortality.

Following enrolment more patients aged 90 years and over died during the study period than presented to hospital, while in younger age groups hospital presentations outnumbered all-cause mortality (figure 1). A similar trend was observed according to increasing clinical comorbidities (figure 2). This is likely to reflect

circumstances in which hospital admission is not in accordance with a patient's wishes or considered clinically appropriate, and end-of-life care is therefore initiated in a person's usual place of residence.

Taken together, our results support the hypothesis that oximetry monitoring is a safe pathway for patients with COVID-19. This was reported to NHS England and Improvement in October 2020.

### Strengths and limitations
The primary strength of this study is its system-level and patient-level impact, since this work evaluated a real-world

**Table 3** Univariable and multivariable binary logistic regression output for presentation to hospital following enrolment to the home oximetry pathway

| | | Univariable | | | Multivariable | | |
| | | OR | P value | 95% CI of OR | OR | P value | 95% CI of OR |
| --- | --- | --- | --- | --- | --- | --- | --- |
| Age | | 1.03 | < 0.001 | 1.01 to 1.05 | 1.03 | 0.018 | 1.00 to 1.05 |
| Ethnicity | White | Reference | | | Reference | | |
| | Black, Asian and minority ethnic | 1.22 | 0.544 | 0.64 to 2.32 | 1.57 | 0.198 | 0.79 to 3.10 |
| | Not recorded | 0.75 | 0.447 | 0.36 to 1.57 | 0.95 | 0.912 | 0.42 to 2.17 |
| Number of comorbidities | | 1.25 | 0.004 | 1.08 to 1.45 | 1.07 | 0.503 | 0.88 to 1.30 |
| Overweight or obese | | 0.67 | 0.175 | 0.38 to 1.19 | 0.55 | 0.069 | 0.29 to 1.05 |
| Clinical severity based on oxygen saturations | Mild (≥95%) | Reference | | | Reference | | |
| | Moderate (93%–94%) | 1.16 | 0.766 | 0.44 to 3.01 | 1.08 | 0.885 | 0.40 to 2.92 |
| | Severe (≤92%) | 2.18 | 0.091 | 0.88 to 5.39 | 2.35 | 0.096 | 0.86 to 6.46 |
| Enrolment pathway | Primary care | Reference | | | Reference | | |
| | Accident and emergency | 0.37 | 0.005 | 0.18 to 0.74 | 0.42 | 0.024 | 0.20 to 0.89 |
| | Inpatient | 0.49 | 0.049 | 0.25 to 0.98 | 0.31 | 0.003 | 0.15 to 0.68 |

**Table 4** Univariable and multivariable binary logistic regression output for all-cause mortality following enrolment to the home oximetry pathway

| | | Univariable | | | Multivariable | | |
|---|---|---|---|---|---|---|---|
| | | OR | P value | 95% CI of OR | OR | P value | 95% CI of OR |
| Age | | 1.12 | <0.001 | 1.08 to 1.16 | 1.08 | <0.001 | 1.03 to 1.13 |
| Ethnicity | White | Reference | | | Reference | | |
| | Black, Asian and minority Ethnic | 0.16 | 0.015 | 0.04 to 0.70 | 0.37 | 0.235 | 0.07 to 1.90 |
| | Not recorded | 0.49 | 0.129 | 0.19 to 1.23 | 0.61 | 0.371 | 0.21 to 1.79 |
| Number of comorbidities | | 1.97 | <0.001 | 1.63 to 2.40 | 1.45 | 0.009 | 1.09 to 1.92 |
| Overweight or obese | | 3.87 | 0.004 | 1.56 to 9.64 | 4.83 | 0.002 | 1.74 to 13.37 |
| Clinical severity based on oxygen saturations | Mild (≥95%) | Reference | | | Reference | | |
| | Moderate (93%–94%) | 2.43 | 0.083 | 0.89 to 6.68 | 1.11 | 0.864 | 0.34 to 3.67 |
| | Severe (≤92%) | 2.16 | 0.224 | 0.62 to 7.53 | 0.65 | 0.574 | 0.14 to 2.93 |
| Enrolment pathway | Primary care | Reference | | | Reference | | |
| | Accident and emergency | 0.66 | 0.587 | 0.15 to 2.97 | 3.40 | 0.157 | 0.62 to 18.55 |
| | Inpatient | 6.57 | 0.001 | 2.23 to 19.41 | 8.70 | 0.001 | 2.53 to 29.89 |

pilot of a new clinical pathway. The project team achieved an integrated partnership between academics, clinicians and policymakers. This enabled a direct pipeline from evidence generation to policy decision-making; moreover, the efficiency with which the work was conducted was nationally important, as it was required to inform how the NHS would use home oximetry monitoring during the next wave of the pandemic. Impact was achieved through the unique composition of the evaluation team as well as the close relationship the team held to NHS leaders in home oximetry monitoring. The method for this work required novel data reporting from the pilot sites to the evaluation team and this was ensured through close collaboration with the individual sites.

The most prevalent limitation related to data quality and completeness. The pilot sites varied in their routes of referral and in the clinical severity of the patients they enrolled. Furthermore, there was heterogeneity in the actual intervention across the sites, as some enrolled patients with suspected COVID-19, while others included patients who had received hospital treatment for COVID-19 and were subsequently discharged. Due to the rapid timescales of this evaluation and the pressured environment in which it took place, mandating full completeness in the data submitted was not possible, and the resulting data set had a considerable amount of missing data. The lack of longitudinal oxygen saturation data precluded temporal analysis of patient trajectories and identification of deterioration in clinical status. While such an evaluation would have been valuable, it was not possible within the available data set. In addition to oxygen saturations, other physiological parameters, including respiratory rate, may be important indicators of clinical severity in patients with COVID-19.[18] In this study, only oxygen saturations were available which precluded

analysis of clinical severity according to a broader range of physiological parameters.

In addition, the absence of precise temporal recording of data and an inability to complete individual case note reviews precluded a more detailed evaluation of whether deaths occurred as a result of COVID-19 infection, or another cause entirely. Finally, this was not a controlled study, which prevented a comparative analysis. Due to these limitations and the relatively low community prevalence of COVID-19 during the pilot, our findings are not necessarily generalisable to future waves of COVID-19. Additionally, while many initial oxygen saturations are likely to be taken in the presence of a healthcare professional, we are unable to determine whether these readings are taken correctly and may therefore provide inappropriate estimates of clinical acuity.

### Comparison to other studies

While research evidence regarding COVID-19 home oximetry is still in its infancy, this study reflects similar findings in the published literature. One systematic review of home monitoring for COVID-19 acknowledges inconclusive evidence about effectiveness and safety. While this study could not conclusively determine the clinical effectiveness of the pilots, it was able to demonstrate non-inferiority in terms of safety with traditional hospital management of COVID-19.

Furthermore, this study indicates the potential for these technologies to support pandemic management in line with expert opinions published in the literature.[19] The findings of this work are consistent with existing grey literature suggesting the utility of home oximetry monitoring for COVID-19.[8 20] This study specifically assessed the safety of home oximetry for COVID-19 and makes policy recommendations not present in existing studies which tend to

focus more on patient experience and potential clinical effectiveness. These findings echo the findings of a previous systematic review indicating a paucity of studies directly addressing patient safety in telemedicine.[21] Identifying the potential patient safety risks arising from the national home oximetry programme is an area of ongoing qualitative and quantitative research as part of the national COVID-19 Oximetry at Home Programme.[20]

### Implications for policy

This work was explicitly designed to answer relevant questions in advance of the national implementation of a new clinical pathway for COVID-19. It confirms the appropriateness of existing policies and that safety netting approaches are congruent with findings.

This study provides information as to how the pathway should function including the thresholds at which people should be enrolled into home oximetry monitoring programmes, how long they should continue in home oximetry monitoring and what sorts of technology are required to support home oximetry monitoring for COVID-19.

Furthermore, the practicalities of running this study revealed important considerations for the ongoing evaluation of home oximetry monitoring pathways for COVID-19. It is clear from the study that it is only possible to measure the safety of home oximetry monitoring with the robust and rigorous collection of data from healthcare providers as well as any third party app-based providers. This data flow should continue during national implementation to continuously measure safety under different conditions and levels of organisational pressure.

### CONCLUSION

Advancing the evidence base regarding the safety of home oximetry monitoring for patients with COVID-19 is of considerable importance as many health systems face new waves of the pandemic worldwide. This study reveals, via a real-world pilot evaluation, that home oximetry monitoring can be a safe pathway for patients with COVID-19; however, substantial research is needed to understand its clinical effectiveness across patient populations. This study was limited by complex and incomplete data as well as variation in intervention designs; a product of the pandemic context within which it was undertaken. However, it did provide initial evidence of the appropriate clinical thresholds and patient characteristics for home oximetry. It also indicated increases in risk to vulnerable groups and patients with oxygen saturations below 95% at enrolment.

Findings from this evaluation have contributed to the national implementation of home oximetry across England, and further work will be undertaken to evaluate clinical effectiveness and any inequalities in terms of access to, and inclusion in, the new pathway.

**Acknowledgements** The research team would like to thank the teams at NHS Digital and NHS England/Improvement as well as the site leads from the home oximetry pilot sites.

**Contributors** JC, KF and RFC were involved in all aspects of the study. HA, GF, JB, AD and SE were involved in the conceptualisation of the study, interpretation of findings and in the reviewing and editing of the draft. JC has had access to all the data in the study and all authors had final responsibility for the decision to submit for publication.

**Funding** This research was supported by the National Institute for Health Research (NIHR) Imperial Patient Safety Translational Research Centre (PSTRC). JC acknowledges support from the Wellcome Trust (215938/Z/19/Z).

**Disclaimer** The views expressed are those of the authors and not necessarily those of the NHS, the NIHR or the Department of Health and Social Care.

**Competing interests** None declared.

**Patient consent for publication** Not required.

**Ethics approval** The work was conducted as a service evaluation, as institutional research governance deemed that it did not require further ethics committee approval. Information governance approval was obtained and a data sharing agreement established (DARS-NIC-396113-N9L4L-v1.2).

**Provenance and peer review** Not commissioned; externally peer reviewed.

**Data availability statement** Data may be obtained from a third party and are not publicly available. Patient-level pseudonymised data were obtained from NHS Digital for the specific purpose of the evaluation. Aggregate findings are presented in this study. Patient-level data may be obtained through application and review by NHS Digital.

**ORCID iDs**
Jonathan Clarke http://orcid.org/0000-0003-1495-7746
Hutan Ashrafian http://orcid.org/0000-0003-1668-0672
Jonathan Benger http://orcid.org/0000-0001-6131-0916

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
