## [Reviewer comments · BMJ Open]

ARTICLE DETAILS

TITLE (PROVISIONAL)	Assessing the Safety of Home Oximetry for Covid-19: A multi-site retrospective observational study.
AUTHORS	Clarke, Jonathan ; Flott, Kelsey; Fernandez Crespo, Roberto; Ashrafian, Hutan; Fontana, Gianluca; Benger, Jonathan; Darzi, Ara; Elkin, Sarah

VERSION 1 – REVIEW

REVIEWER	Jouffroy, Romain Assistance Publique - Hopitaux de Paris, Intensive Care Unit, Anesthesiology, SAMU, Necker - Enfants Malades Hospital
REVIEW RETURNED	05-Mar-2021

GENERAL COMMENTS	Thank you for allowing me to review the manuscript entitled “Assessing the Safety of Home Oximetry for Covid-19: A multi-site retrospective observational study.”. The current article reports the the safety and effectiveness of home oximetry monitoring pathways safe for Covid-19 patients in the English NHS. The text is well written. I have comments on the form and on the background. Form - Abstract: Objectives: please add a dot at the end of the sentence- Abstract: please 95 per cent confidence interval for each Odd Ratio- References format overall the manuscript: prefer, for example (1-3) instead of (1) (2) (3) when more than 2 references Background Are the devices used for oxygen saturation assessment, correctly used by patients and validated against reference material? In my opinion, 2 issues impact your results: 1. The main outcome is perhaps to ambitious? This results in a restricted number of patients with the main outcome. It’s difficult to believe that only one element directly impacts the mortality which is a dimension affected by multiple factors (age, underlying comorbidities for example). Consequently, the restricted number of patients with outcome limits the statistical analysis2. It would be more interested to analyse the trends of pulse oximetry variation than a single assessment
---

REVIEWER	Irving, Greg University of Cambridge
REVIEW RETURNED	23-Mar-2021

GENERAL COMMENTS	* This paper addresses an important research question in relation to patient safety at a crucial time when many areas in the UK are making future commissioning decision in relation to COVID Oximetry services. *It may be helpful to emphasise the scale of implementation of COVID Oximetry across the NHS in England e.g. at the time of writing over twenty seven thousand high risk patients with COVID-19 have been monitored across all CCGs. * The authors may wish to consider highlighting that patient safety issues related telehealth can occur at a number of levels and that historically there has been a paucity of studies specifically designed to address this e.g. Guise et al. * The methods used are appropriate for the stated research question. The statistical methods used are appropriate and analysis correct - you may wish for a statistician to review this in more detail. *More detail of the Oximetry service itself would've been helpful – perhaps following the TIDieR framework and / or including the Oximetry standard operating procedure in the appendix. Services can vary across CCGs. * In Table 1 of the base line characteristics of the patients included it would've been helpful to see other COVID risk factors described e.g. learning disability and the QCOVID score itself.
---

VERSION 1 – AUTHOR RESPONSE

Reviewer: 1

Dr. Romain Jouffroy, Assistance Publique - Hopitaux de Paris Comments to the Author:

Form

- Abstract: Objectives: please add a dot at the end of the sentence
- Abstract: please 95 per cent confidence interval for each Odd Ratio
- References format overall the manuscript: prefer, for example (1-3) instead of (1) (2) (3) when more than 2 references

Thank you for these suggestions. We have made the appropriate changes to the text.

Background

Are the devices used for oxygen saturation assessment, correctly used by patients and validated against reference material?

It is not possible to ascertain whether the use of oximeters by individual patients was correct at all times. At initiation of the service, baseline measurements are taken in the presence of a health professional and it is these first measurements that are used in this study. It is not possible to ascertain consistent correct use of oximeters while on the pathway, however these do not form part of this study. We have added this important consideration to the limitations section of the manuscript.

In my opinion, 2 issues impact your results:

1. The main outcome is perhaps too ambitious? This results in a restricted number of patients with the main outcome. It's difficult to believe that only one element directly impacts the mortality which is a dimension affected by multiple factors (age, underlying comorbidities for example). Consequently, the restricted number of patients with outcome limits the statistical analysis

Thank you. We agree that there are many factors influencing mortality in those enrolled on home monitoring pathways. For this reason we aimed to include a wide range of clinical risk factors from primary care data. We also sought to include those with available oxygen saturation readings due to the significant relationship this has with clinical severity in Covid-19. While this restricts the numbers included in the study, it still represents the largest cohort study of home oximetry pathways in England and is able to offer important insights into the behaviour of different types of clinical pathways included.

2. It would be more interesting to analyse the trends of pulse oximetry variation than a single assessment

We agree that examining trends in oxygen saturations would be helpful to identify deterioration or improvement while on the pathway. Unfortunately, these data were not available with sufficient consistency to enable longitudinal analysis. We have added a comment to the study limitations to discuss this.

Reviewer: 2

Dr. Greg Irving, University of Cambridge Comments to the Author:

* This paper addresses an important research question in relation to patient safety at a crucial time when many areas in the UK are making future commissioning decisions in relation to COVID Oximetry services.

Thank you for your supportive review and helpful comments. We have addressed each in turn below.

*It may be helpful to emphasise the scale of implementation of COVID Oximetry across the NHS in England e.g. at the time of writing over twenty seven thousand high risk patients with COVID-19 have been monitored across all CCGs.

Thank you for this. At the time of writing the initial submission (December 2020) the Covid Oximetry at Home pathway was rapidly expanding across England and we have added a statement to the introduction to reflect the estimated total number of patients who have used home oximetry pathways.

* The authors may wish to consider highlighting that patient safety issues related to telehealth can occur at a number of levels and that historically there has been a paucity of studies specifically designed to address this e.g. Guise et al.

Thank you for raising this important point. Identifying the potential patient safety risks arising from the national home oximetry programme is an area of ongoing qualitative and quantitative research. We have raised this in the discussion.

* The methods used are appropriate for the stated research question. The statistical methods used are appropriate and analysis correct - you may wish for a statistician to review this in more detail.

Thank you

*More detail of the Oximetry service itself would've been helpful – perhaps following the TIDieR framework and / or including the Oximetry standard operating procedure in the appendix. Services can vary across CCGs.

Thank you for this. We have added a section to the introduction to better describe the intervention using the TIDieR framework as a basis. We have additionally provided the current NHS England SOP as a supplement (<https://www.england.nhs.uk/coronavirus/wp-content/uploads/sites/52/2020/11/C0817-standard-operating-procedure-covid-oximetry-@home-v1.1-march-21.pdf>)

* In Table 1 of the base line characteristics of the patients included it would've been helpful to see other COVID risk factors described e.g. learning disability and the QCOVID score itself.

The QCOVID score was not available in the dataset released from NHS Digital for the purpose of the evaluation. We have included a breakdown of the number of patients with each included clinical risk factor across each route of enrolment, and overall as an extension of Table 1.

VERSION 2 – REVIEW

REVIEWER	Jouffroy, Romain Assistance Publique - Hopitaux de Paris, Intensive Care Unit, Anesthesiology, SAMU, Necker - Enfants Malades Hospital
REVIEW RETURNED	20-May-2021
GENERAL COMMENTS	Thank you for allowing me to review the revised version of manuscript entitled “Assessing the Safety of Home Oximetry for Covid-19: A multi-site retrospective observational study.”. The current article reports the the safety and effectiveness of home oximetry monitoring pathways safe for Covid-19 patients in the English NHS. The text is well written. I have comments on the form and on the background and hope it would be helpful to improve the quality of the manuscript. Form: Abstract: 1338 and 908 in full words because sentence beginning Background We still do not know if the devices used for oxygen saturation assessment were correctly used by patients and validated against reference material. However, this limit was inserted in the revised version of the manuscript. Independently, of pulse oximetry

values, the authors could discuss the helpfulness of respiratory rate to assess severity, see reference: PMID: 32513249

VERSION 2 – AUTHOR’S RESPONSE

Thank you for your helpful suggestions to the revised manuscript. We have adopted the suggestions made in the editor's note regarding sentences beginning with numbers. In order to remain within the word count for the abstract we made other small changes to the abstract wording. We hope this is satisfactory.

Thank you for raising the important role other physiological parameters, particularly respiratory rate, in determining clinical acuity for Covid-19 patients. We have added to the limitations section of the manuscript to reflect the absence of measures outside of oxygen saturations in our study and have included the helpful reference in support of inclusion of other parameters.